# White Collar 1 Modulates Oxidative Sensitivity and Virulence by Regulating the HOG1 Pathway in *Fusarium asiaticum*

Ying Tang,[a] Yan Tang,[a] Dandan Ren,[a] Congcong Wang,[a] Yao Qu,[a,b] Li Huang,[a,c] Yongjun Xue,[a] Yina Jiang,[a] Yiwen Wang,[a] Ling Xu,[a] ⓘ Pinkuan Zhu[a]

aSchool of Life Sciences, East China Normal University, Shanghai, China
bNo. 2 High School of East China Normal University, Shanghai, China
cSuzhou Industrial Park Xingyang School, Suzhou, China

Ying Tang and Yan Tang contributed equally to this paper. Author order was determined in order of decreasing stroke numbers of their names written in Chinese.

**ABSTRACT** *Fusarium asiaticum* is an epidemiologically important pathogen of cereal crops in east Asia, accounting for both yield losses and mycotoxin contamination problems in food and feed products. FaWC1, a component of the blue-light receptor White Collar complex (WCC), relies on its transcriptional regulatory zinc finger domain rather than the light-oxygen-voltage domain to regulate pathogenicity of *F. asiaticum*, although the downstream mechanisms remain obscure. In this study, the pathogenicity factors regulated by FaWC1 were analyzed. It was found that loss of FaWC1 resulted in higher sensitivity to reactive oxygen species (ROS) than in the wild type, while exogenous application of the ROS quencher ascorbic acid restored the pathogenicity of the Δ*Fawc1* strain to the level of the wild type, indicating that the reduced pathogenicity of the Δ*Fawc1* strain is due to a defect in ROS tolerance. Moreover, the expression levels of the high-osmolarity glycerol (HOG) mitogen-activated protein kinase (MAPK) pathway genes and their downstream genes encoding ROS scavenging enzymes were downregulated in the Δ*Fawc1* mutant. Upon ROS stimulation, the FaHOG1-green fluorescent protein (GFP)-expressing signal driven by the native promoter was inducible in the wild type but negligible in the Δ*Fawc1* strain. Overexpressing *Fahog1* in the Δ*Fawc1* strain could recover the ROS tolerance and pathogenicity of the Δ*Fawc1* mutant, but it remained defective in light responsiveness. In summary, this study dissected the roles of the blue-light receptor component FaWC1 in regulating expression levels of the intracellular HOG-MAPK signaling pathway to affect ROS sensitivity and pathogenicity in *F. asiaticum*.

**IMPORTANCE** The well-conserved fungal blue-light receptor White Collar complex (WCC) is known to regulate virulence of several pathogenic species for either plant or human hosts, but how WCC determines fungal pathogenicity remains largely unknown. The WCC component FaWC1 in the cereal pathogen *Fusarium asiaticum* was previously found to be required for full virulence. The present study dissected the roles of FaWC1 in regulating the intracellular HOG MAPK signaling pathway to affect ROS sensitivity and pathogenicity in *F. asiaticum*. This work thus extends knowledge of the association between fungal light receptors and the intracellular stress signaling pathway to regulate oxidative stress tolerance and pathogenicity in an epidemiologically important fungal pathogen of cereal crops.

**KEYWORDS** light receptor, MAPK, pathogenicity, ROS, transcription factor

Light is a common environmental factor that drives photosynthesis and determines the behaviors of living organisms to adapt to the niches they live in (1). Fungi are eukaryotic, heterotrophic, and sessile organisms that can be either friends or foes: they can be decomposers in nature, helping to maintain circulation balance of materials in

Address correspondence to Ling Xu, lxu@bio.ecnu.edu.cn, or Pinkuan Zhu, pkzhu@bio.ecnu.edu.cn.

The authors declare no conflict of interest.

the world, or metabolite producers for the food and medicine industries, but they can also be devastating pathogens of plants and animals (2). The presence of light usually tells fungi if they are potentially being exposed to high temperature or to harmful irradiation with associated DNA damage, accumulation of reactive oxygen species (ROS), desiccation, and other stressful conditions (3, 4). Thus, understanding the light signaling mechanisms in fungi may provide benefits for pathogen control and/or biotechnological processes.

Light sensing in fungi can be achieved directly via photoreceptor proteins. The molecular mechanisms of fungal light responses have been studied best in *Neurospora crassa*, whose light response phenomena have been known for decades, and the isolation of light-insensitive mutants finally led to identification of the White Collar complex (WCC) photoreceptor (5, 6). WCC is a heterodimer composed of WC-1 and WC-2, which are both transcriptional regulators; they can directly bind to the promoters of light-activated genes and, after illumination, activate their expression (6, 7). Therefore, a complex signal transduction cascade is likely unnecessary for WCC to transmit the light signal to initiate differential gene expression. Many of the early light-responsive target genes of WCC encode transcription factors. These light-regulated transcription factors together with the WCC further activate downstream genes that are closely related to light responses like hyphal growth, development, metabolisms, asexual/sexual reproduction, UV damage repair, phototropism, circadian rhythms, etc. (8, 9). Moreover, the WCC-regulated target genes could also encode some WCC inhibitors, such as another blue-light receptor, vivid (VVD), which is highly expressed upon light exposure and interacts with the WCC to function as a negative regulator of WCC activity for photoadaptation (10, 11).

Although the molecular components for blue-light sensing appear to be widely conserved in fungal genomes, the study of light regulation in an increasing number of fungal organisms has revealed a high complexity, and the regulatory circuits and the sensitivity of certain species to specific wavelengths are different (3, 4). There are characterized fungal photoreceptors other than WCC, including phytochromes, opsins, and cryptochromes that are responsible for sensing red/far-red, green, and UV light qualities, respectively, and the progress in understanding photobiology in both models and other fungal species has been well summarized in extensive reviews (3, 4, 12–14). It is worth noting that fungal photoreceptors have been shown to be involved in regulating virulence of certain pathogenic fungi in either light-dependent or -independent manners (15–19). However, the downstream targets that are regulated by fungal photoreceptors and are related to virulence determinants remain unknown. Moreover, signaling output target genes of the fungal photoreceptors can be species specific and are considered the result of adaptive evolution; signaling cascades may trigger different responses even in closely related species, highlighting the need to study their functions in specific organisms (3).

Besides photoreceptors, fungi can sense light indirectly through perceiving the fluctuation of ROS when light is absorbed by intracellular photosensitizers, which were tested in the budding yeast *Saccharomyces cerevisiae* lacking photoreceptors (20, 21). In line with the idea that light exposure generates ROS, proteins of the oxidative stress response were associated with responses to light (12). The high-osmolarity glycerol (HOG) mitogen-activated protein kinase (MAPK) plays a central role in responding to oxidative stress in *S. cerevisiae*, and thus, HOG pathway signaling is likely required for light sensing in yeast (20). Functional HOG pathways have been identified to be conserved in many other fungi, and this pathway is used by fungi to adapt to stress conditions other than osmotic stress; therefore, the HOG kinase gene was named *stress-activated kinase A* (*sakA*) in *Aspergillus nidulans* (22). In fungal pathogens, the invasion processes inevitably suffer from the stresses exerted by host defense (or immunity) responses, among which the most typical response is oxidative stress caused by an ROS burst in the infected host. Consequently, successful pathogens should be able to cope with such hyperoxidative stress during interaction with their hosts, while the HOG pathway is commonly required for host colonization by pathogenic fungi mainly due to its vital role in fungal adaptation to ROS stress (e.g.,

*Magnaporthe grisea* and *Cryptococcus neoformans*) (23, 24). Intriguingly, the HOG pathway is also essential for the phytochrome-mediated light sensing in *A. nidulans* and *Alternaria alternata* (25, 26). So, the questions of whether and how the fungal light signaling components could be associated with the stress response pathway to regulate pathogenicity deserve further research efforts.

The genus *Fusarium* comprises a large group of ubiquitous fungal species, which are frequently associated with mycotoxin production and plant pathogenesis of great economic importance (27). The *Fusarium* species also stand out as research models for fungal photobiology due to their obvious light-sensing phenotypes, which include asexual and sexual reproduction, secondary metabolites, and stress tolerance (28). The *Fusarium graminearum* species complex (FGSC) is the predominant set of pathogens for *Fusarium* head blight (FHB) on cereal crops, including wheat, barley, and other cereal grains all over the world (29). The FHB cause not only quantitative yield losses in production but also qualitative problems of contamination with trichothecene mycotoxins in the harvested crops. These toxin metabolites pose a significant risk to food and feed safety because they inhibit eukaryotic protein synthesis and modify immune function. In east Asia, the FHB-causing strains mainly consist of two groups, *Fusarium graminearum* sensu stricto and *Fusarium asiaticum* (30–32). *F. graminearum* sensu stricto commonly exists in cooler regions, while the vast majority of *F. asiaticum* isolates have been found in warmer regions (30, 32). Light-sensing responses have been reported in both *F. graminearum* and *F. asiaticum*, and the blue-light receptor WCC was found to be responsible for regulating their growth, development, and metabolism responses to light stimulus (33, 34). Moreover, the WCC component FaWC1 of *F. asiaticum* demonstrated a light-independent role in regulating virulence via its zinc finger domain but not its light, oxygen, or voltage (LOV) domain, suggesting that it is the transcription factor rather than its light-sensing function that is required for virulence (34), although the molecular mechanisms that allow this photoreceptor to specifically regulate virulence remain elusive.

The present study aimed to identify the potential targets of *F. asiaticum* WC1 (FaWC1). The hypersensitivity of a Δ*Fawc1* mutant to ROS was identified and associated with its virulence deficiency. The stress tolerance-related HOG pathway genes and ROS-scavenging enzymes were further determined to be regulated by FaWC1. Altogether, the data in this report show that the WCC component FaWC1 of *F. asiaticum* can regulate the HOG pathway to affect ROS sensitivity and pathogenicity, thus extending our understanding of how the conserved WC1 is interlinked with intracellular stress signaling.

## RESULTS

**FaWC1 has a role in oxidative stress tolerance.** Our earlier study demonstrated that FaWC1 plays a vital role in regulating virulence of *F. asiaticum* (34). As tolerance for oxidative stress can be activated under light irradiation and during host invasion, this study subsequently measured the sensitivities of *F. asiaticum* wild-type (WT) and mutant strains to constant blue light and the oxidative agent $H_2O_2$. The colony expansion rates were suppressed by constant blue light to a greater extent in the Δ*Fawc1* mutant than in the WT strain (Fig. 1A and B), indicating that loss of FaWC1 resulted in defective tolerance to blue-light irradiance stress. As demonstrated by spore germination and colony growth assays, both the Δ*Fawc1* and Δ*Fawc1*-C$^{\Delta ZnF}$ mutants were more sensitive to $H_2O_2$ than the WT, Δ*Fawc1*-C, and Δ*Fawc1*-C$^{\Delta LOV}$ strains (Fig. 1C to E). Moreover, 3,3′-diaminobenzidine (DAB) staining with the mycelial colonies demonstrated that the Δ*Fawc1* and Δ*Fawc1*-C$^{\Delta ZnF}$ mutants accumulated markedly stronger ROS than WT, Δ*Fawc1*-C, and Δ*Fawc1*-C$^{\Delta LOV}$ strains (Fig. 2A). These results suggest that FaWC1 positively regulates tolerance to photooxidative stress of constant blue light, and disruption of the whole FaWC1 or partial deletion of the ZnF domain of FaWC1 could lead to elevated sensitivity to ROS, probably due to defects in ROS-scavenging abilities.

Since host invasion by the pathogenic fungi should also confront oxidative burst stress, wheat coleoptiles were artificially inoculated to test the effect of the ROS-quenching agent ascorbic acid on disease severity caused by the WT and Δ*Fawc1*

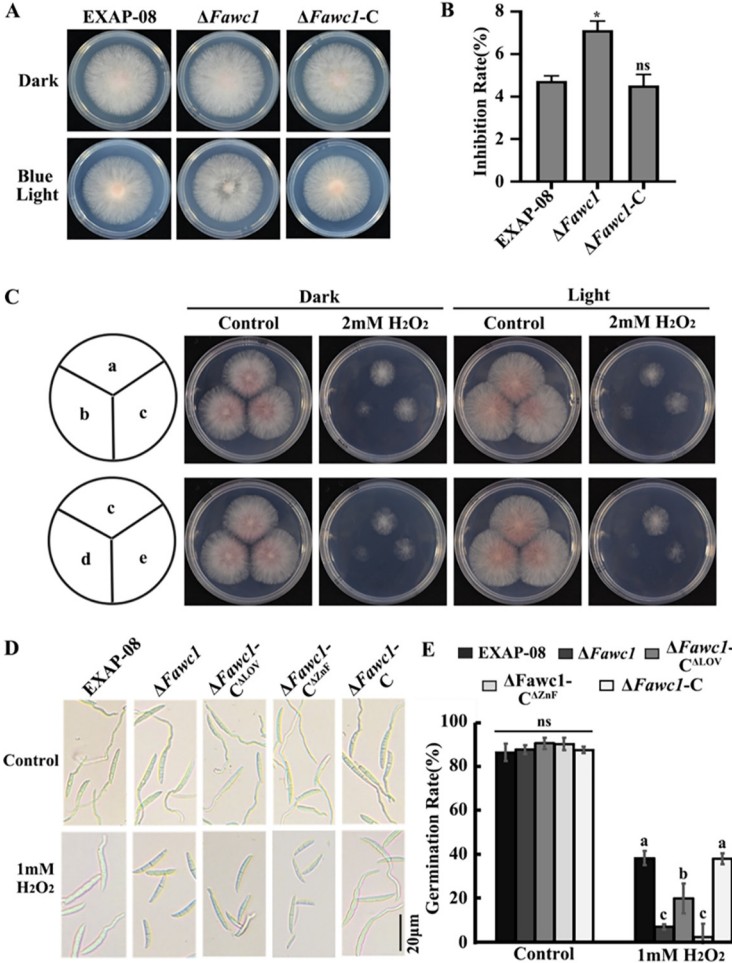

**FIG 1** FaWC1 is required for tolerance to oxidative stresses. (A and B) Colony expansion rates were suppressed by constant blue light to greater extents in the Δ*Fawc1* mutant than in the WT (EXAP-08) and Δ*Fawc1*-C strains. Cultures were grown in either constant darkness or blue light for 2 days. (C) Mycelium growth of the Δ*Fawc1* and Δ*Fawc1*-C$^{\Delta ZnF}$ mutants was more sensitive to H$_2$O$_2$ than that of the WT, Δ*Fawc1*-C, and Δ*Fawc1*-C$^{\Delta LOV}$ strains. The letter (a to e) in the pie charts represent inoculation sites of WT, Δ*Fawc1*, Δ*Fawc1*-C, Δ*Fawc1*-C$^{\Delta ZnF}$, and Δ*Fawc1*-C$^{\Delta LOV}$ strains, respectively, in the petri dishes. (D and E) Spore germination rates of the Δ*Fawc1* and Δ*Fawc1*-C$^{\Delta ZnF}$ mutants were more strongly inhibited by 1 mM H$_2$O$_2$ than those of the WT, Δ*Fawc1*-C, and Δ*Fawc1*-C$^{\Delta LOV}$ strains. Bar, 50 $\mu$m. The columns marked with different letters are significantly different from each other at a *P* value of <0.05. ns, no significant difference among the compared samples.

strains. In the control coleoptile samples, infection with the WT caused larger decaying areas than infection with the Δ*Fawc1* mutant. However, ascorbic acid treatment recovered the disease severity in the Δ*Fawc1* mutant-infected coleoptiles to the level of the WT-infected samples (Fig. 2B and C). Overall, this inoculation assay further suggests that the attenuated virulence of the Δ*Fawc1* strain can be likely ascribed to its defects in ROS tolerance capacity.

**Nucleus localization activity of FaWC1 is regulated by both light and oxidative stimuli.** Considering that FaWC1 is involved in regulating tolerance to oxidative stresses, while the White Collar photoreceptor is also a transcription factor, the subsequent assays analyzed dynamics of the FaWC1 subcellular localization using the Δ*Fawc1*::FaWC1-GFP strain, which expressed the FaWC1-green fluorescent protein (GFP) fusion protein in the Δ*Fawc1* background and complemented the light response phenotypes to WT levels (Fig. 3A and B). In the germinating asexual spores, FaWC1-GFP mainly demonstrated cytosolic distribution under constant darkness (2 h D) and light (2 h L) conditions. In contrast, 5 min light illumination after dark incubation (2 h D + 5 min L) induced an obvious accumulation of the FaWC1-GFP signals in nuclei.

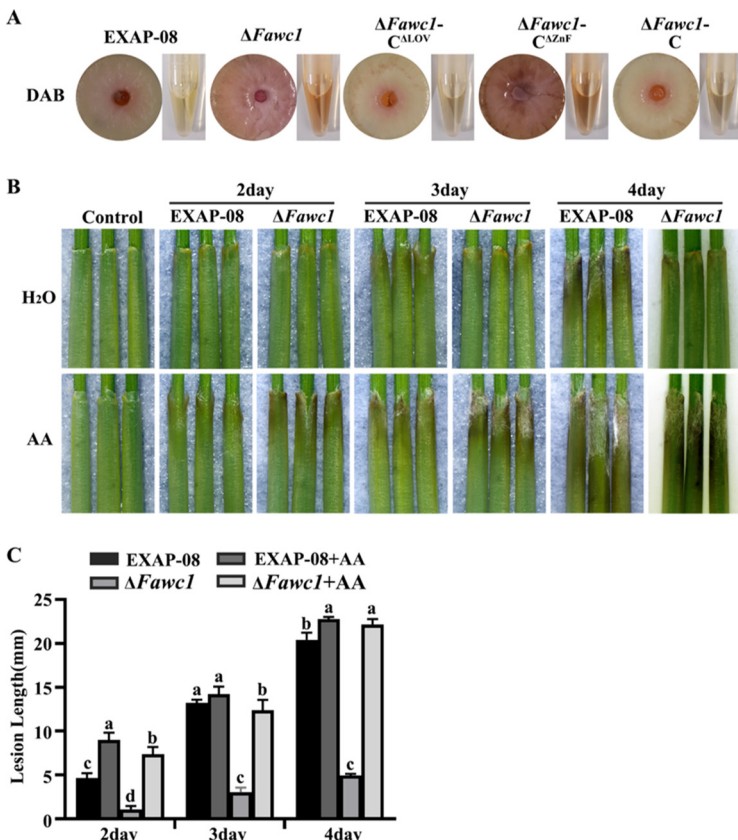

**FIG 2** FaWC1 is indispensable for scavenging ROS during growth *in vitro* and pathogenicity *in vivo*. (A) Cultures of Δ*Fawc1* and Δ*Fawc1*-C$^{\Delta ZnF}$ mutants accumulated more ROS, as indicated by DAB staining, than those of the WT, Δ*Fawc1*-C, and Δ*Fawc1*-C$^{\Delta LOV}$ strains. (B) Wheat coleoptiles infected with the WT and Δ*Fawc1* strains were photographed 3 days after inoculation. (C) The pathogenicity defects of the Δ*Fawc1* mutant could be recovered to levels comparable to those of the WT strain by exogenous application of the ROS-quenching agent ascorbic acid (AA). Columns marked with different letters are significantly different from each other at a *P* value of <0.05.

However, as the illumination continued, as exemplified with 2 h D + 10 min L and 2 h D + 15 min L treatments, the nuclear accumulation of FaWC1-GFP signals gradually decreased. Intriguingly, $H_2O_2$ treatment also led to apparent accumulation of the FaWC1-GFP in nuclei (Fig. 3C and D). Collectively, these data suggest that FaWC1 responds to light by translocating quickly from the cytoplasm to the nucleus; meanwhile, its subcellular localization in the nucleus is also sensitive to oxidative stimuli.

**Expression levels of the genes related to ROS scavenging are regulated by FaWC1.** To test the hypothesis that FaWC1 regulates the ROS scavenging capacity of *F. asiaticum*, the next assays measured the transcript levels of the genes related to ROS degradation activities. The expression levels of *Fakatg2*, *Facat4/6*, and *Fasod* genes, which encode catalase-peroxidase, catalases, and superoxide dismutase, respectively, were enhanced by $H_2O_2$ treatment in the WT strain. However, the *Fakatg2* and *Facat4* expression levels in the Δ*Fawc1* mutant strain under both control and $H_2O_2$ treatment conditions remained comparable to those in the control sample of the WT strain. Additionally, *Facat6* demonstrated a higher expression level in the Δ*Fawc1* mutant than the WT under control conditions, but $H_2O_2$ treatment failed to increase the *Facat6* expression level in the Δ*Fawc1* strain. The transcript level of *Fasod* could be slightly up-regulated by $H_2O_2$ treatment in the Δ*Fawc1* strain, but the increase was less than that in the WT. In general, the Δ*Fawc1* mutant is defective in inducing the expression of the four ROS degradation-related genes upon $H_2O_2$ stimulation (Fig. 4).

In filamentous fungi, these ROS degradation-related genes could be regulated via the conserved high-osmolarity glycerol (HOG) MAP kinase pathway. Subsequent assays

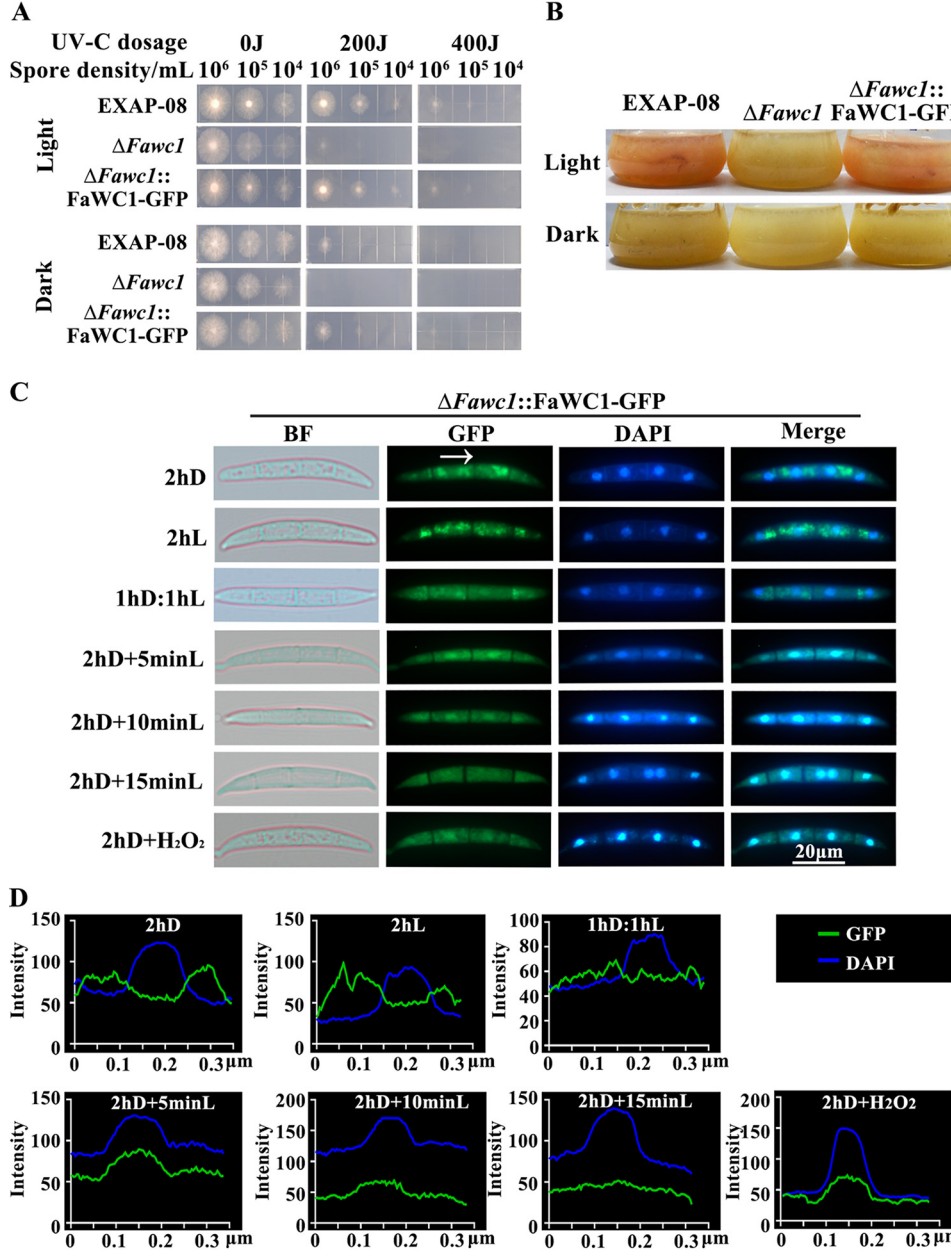

**FIG 3** Nucleus localization of the FaWC1 is responsive to both light and oxidative stimuli. (A) Serially diluted conidial suspensions of the WT, ΔFawc1, and ΔFawc1::FaWC1-GFP strains were applied dropwise to solid CM plates, followed by UV radiation and subsequent culture in either light or darkness to test UV damage tolerance. (B) The three strains were cultured in liquid CM under light or dark conditions to examine the photopigmentation response. The defects of ΔFawc1 in tolerance to light-induced UV damage (A) and pigment accumulation (B) were rescued by transgenic expression of the FaWC1-GFP fusion protein. (C) Subcellular localization of FaWC1-GFP signals could be affected by transient light and H₂O₂ treatment. The conidial suspension of the ΔFawc1::FaWC1-GFP transgenic strain was applied dropwise to glass slides. After incubation in darkness or light for 2 h or in darkness for 2 h followed by light illumination for the indicated times, the conidial samples were stained with DAPI and subsequently observed with a fluorescence microscope under the bright field, GFP, and DAPI channels. Bar, 20 μm. (D) Line scan graphs were generated at the indicated positions to show the relative localization of GFP (green) and DAPI (blue) signals.

analyzed the transcript levels of the HOG pathway genes. The results indicate that transcription levels of *Fassk1*, *Fassk2*, *Fapbs2*, *Fahog1*, and *Faatf1*, which encode homologues of the yeast SSK1 two-component response regulator, MAP3K, MAP2K, HOG1 MAPK, and the downstream Atf1 bZIP transcription factor, respectively, were significantly induced by H₂O₂ treatment in the WT strain. However, expression of these HOG

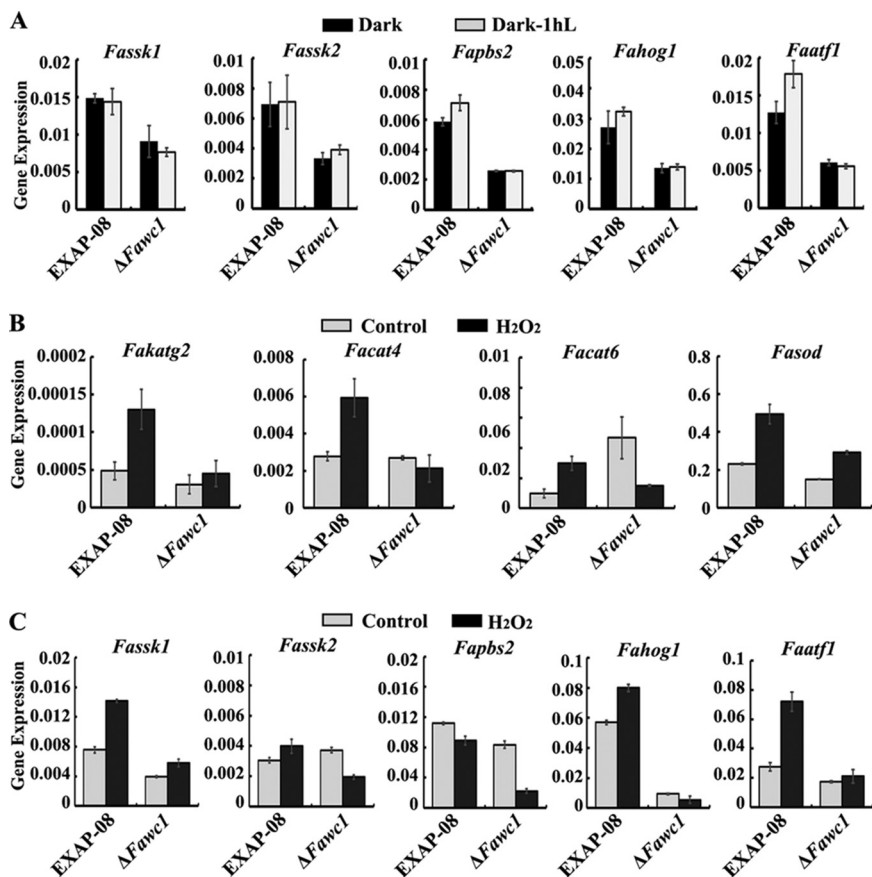

**FIG 4** Expression levels of the genes related to ROS scavenging enzymes and regulating pathway were attenuated in the Δ*Fawc1* mutant. Cultures of WT and Δ*Fawc1* strains were treated with either light illumination or $H_2O_2$. After incubation for 24 h, the mycelium samples were collected and subjected to RNA extraction and qRT-PCR assays to measure the expression levels of ROS-scavenging enzyme genes and HOG MAPK pathway genes. (A) Expression levels of the HOG MAPK pathway genes in response to light treatment. (B and C) Expression levels of ROS-scavenging enzyme genes (B) and HOG MAPK pathway genes (C) upon oxidative stimulation.

pathway genes was not affected by light treatment and remained at low levels upon $H_2O_2$ stimulus in the Δ*Fawc1* mutant (Fig. 4). These data suggest that the defects of the Δ*Fawc1* mutant in oxidative tolerance are likely due to low expression levels of the ROS scavenging enzymes and the HOG signaling pathway genes.

**Disruption of altered FaWC1 expression and subcellular localization patterns of FaHOG1.** The HOG1 MAPK is the vital component of the HOG pathway for regulating stress responses in fungi. The next study additionally generated the Δ*Fahog1* mutant, and the vector for expressing the FaHOG1-GFP fusion protein with the native promoter was transformed into the Δ*Fahog1* strain, resulting in the Δ*Fahog1*::FaHOG1-GFP strain. As expected, the Δ*Fahog1* mutant demonstrated strong defects in tolerance to osmolarity and oxidative stresses, as exerted via 0.5 M salt solutions (NaCl or KCl) and 2.5 mM $H_2O_2$, respectively (Fig. 5). Meanwhile, the Δ*Fahog1*::FaHOG1-GFP (or Δ*Fahog1*-C) strain demonstrated tolerance to the osmolarity and oxidative stresses comparable to those of the WT strain (Fig. 5), suggesting that the FaHOG1-GFP fusion protein was able to recover the defects of the Δ*Fahog1* mutant.

Additionally, the same FaHOG1-GFP fusion protein-expressing vector was transformed into the Δ*Fawc1* strain to generate the Δ*Fawc1*::FaHOG1-GFP strain. Subsequent assays analyzed the dynamic subcellular locations of FaHOG1-GFP in response to various stimuli in the Δ*Fahog1*::FaHOG1-GFP and Δ*Fawc1*::FaHOG1-GFP strains. The FaHOG1-GFP signal was constantly detectable, while light (15 min), oxidant ($H_2O_2$), and high-osmolarity (0.5 M NaCl) stimuli could apparently enhance its localization in nuclei of the germinating spores

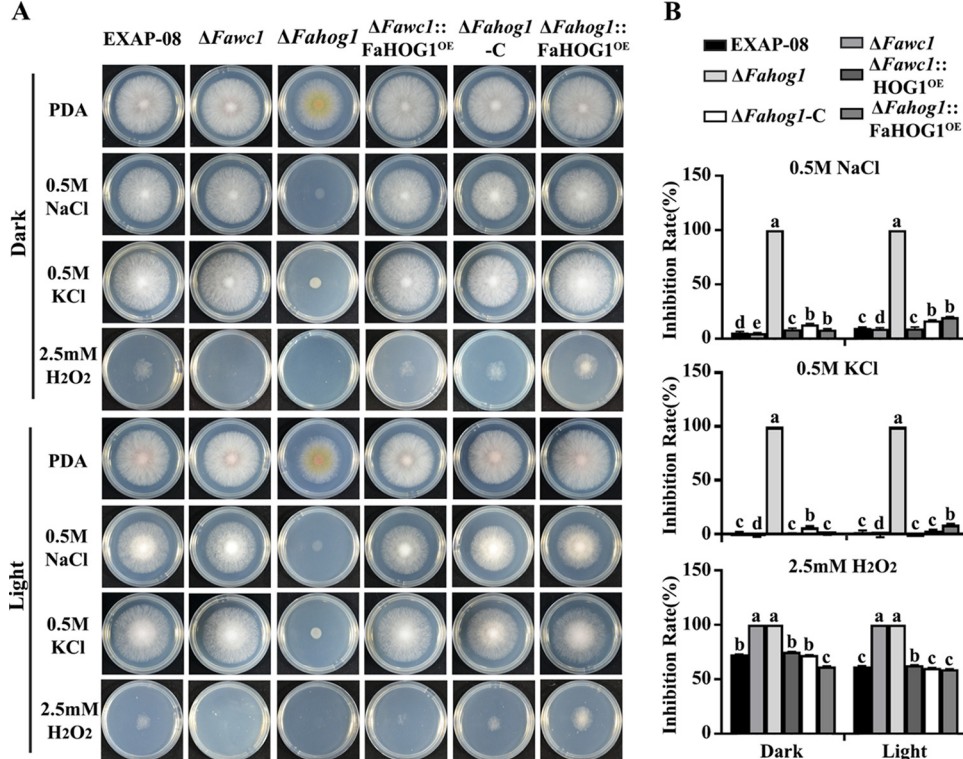

**FIG 5** Overexpression of FaHOG1 rescues the defects of ΔFawc1 in ROS tolerance. (A) The defects of the ΔFahog1 in colony growth under osmotic and oxidative stresses was recovered by overexpressing FaHOG1 in the ΔFahog1::FaHOG1$^{OE}$ transgenic line. The defect of ΔFawc1 in oxidative tolerance as indicated by colony growth was also recovered by overexpressing FaHOG1 in the ΔFawc1::FaHOG1$^{OE}$ transgenic line. (B) Statistic analysis of the colony growth inhibition rates caused by either osmotic or oxidative stresses in the test strains. The columns marked with different letters are significantly different from each other at a P value of <0.05.

of the ΔFahog1::FaHOG1-GFP strain. In contrast, the HOG1-GFP signal in the ΔFawc1::FaHOG1-GFP strain remained almost undetectable under all the conditions tested, with the exception that high osmolarity (0.5 M NaCl) still caused noticeable fluorescent signals in nuclei (Fig. 6; also, see Fig. S2 in the supplemental material). Therefore, the promoter regions of Fahog1 and Fassk2 were analyzed, and two predicted cis-element binding motifs (TCTTCCTCCTC and CCATCTAT) of FaWC1 were found. Subsequently, an electrophoretic mobility shift assay (EMSA) was carried out and confirmed that FaWC1 could bind to the promoter regions of FaHog1 but not Fassk2 (Fig. 7). These data led to the assumption that the expression of FaHog1 could be directly regulated by FaWC1.

Additionally, the critical factor for the HOG1 MAP kinase activity is its phosphorylation level. A Western blot assay subsequently demonstrated that the phosphorylation level of FaHOG1 could be induced by light and H$_2$O$_2$ stress stimuli in the WT strain. In contrast, the FaHOG1 phosphorylation level of the ΔFawc1 strain was not sensitive to light and H$_2$O$_2$ treatments (Fig. 8). Unexpectedly, we did not find a significant change in phosphorylation level of FaHOG1 in response to osmotic stress in either the WT or ΔFawc1 strain, probably because phosphorylation of FaHOG1 upon osmotic stimulus can be so transient that we failed to detect it. In any case, these data collectively revealed that the expression, phosphorylation, and nuclear localization of functional FaHOG1 in response to an oxidative stimulus can be dependent on the presence of FaWC1.

**Overexpression of FaHOG1 rescues the defects of ΔFawc1 in ROS tolerance but not in light sensing.** As FaWC1 is required for light sensing, oxidative stress tolerance, and virulence of *F. asiaticum*, the following assays aimed to test whether the altered Fahog1 expression pattern in the ΔFawc1 mutant was associated with the mutant phenotypes. For this purpose, the Fahog1 overexpression vector was transformed into the

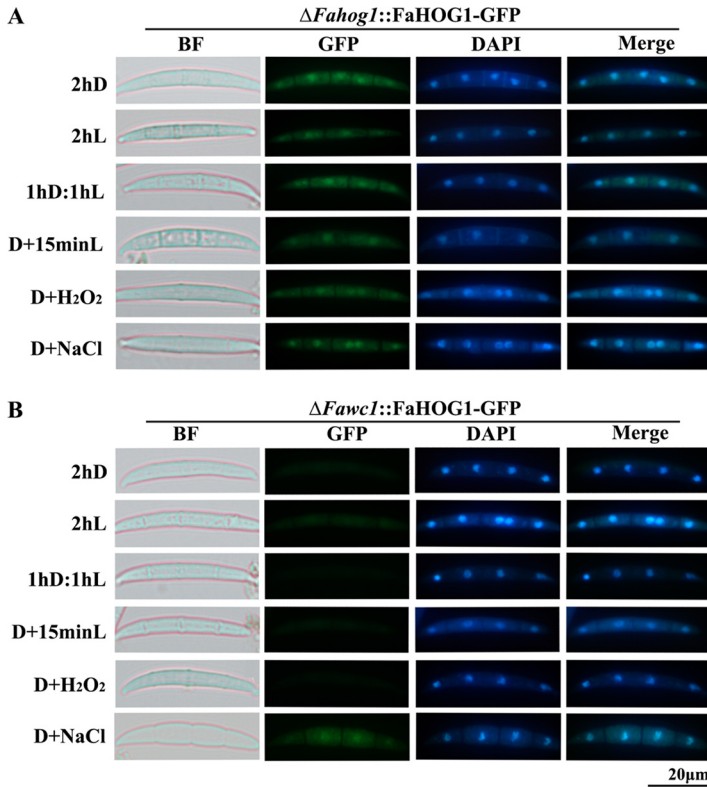

**FIG 6** Expression and subcellular localization of the functional FaHOG1 in response to light and oxidative stimuli are dependent on the presence of FaWC1. (A) The expressed FaHOG1-GFP signals could be accumulated in nuclei upon light, oxidative, and salt stress. (B) Expression and nucleus localization of FaHOG1 induced by light and oxidative stimuli are dependent on the presence of FaWC1, while the osmosis-induced expression and nucleus localization of FaHOG1 are independent of FaWC1. The conidial suspensions of the Δ*Fahog1*::FaHOG1-GFP and Δ*Fawc1*::FaHOG1-GFP transgenic strains incubated on glass slides were treated with light (L), salt (NaCl, 0.5 M), and $H_2O_2$ (2.5 mM). After staining with DAPI, the samples were subsequently observed with a fluorescence microscope under bright-field, GFP, and DAPI channels. Bar, 20 $\mu$m.

Δ*Fawc1* and Δ*Fahog1* mutants, resulting in Δ*Fawc1*::FaHOG1$^{OE}$ and Δ*Fahog1*::FaHOG1$^{OE}$ transgenic strains, respectively. Interestingly, FaHOG1 was vital for resistance to oxidative stress, and the Δ*Fawc1*::FaHOG1$^{OE}$ transgenic strain as the wild type strain to the oxidative agent $H_2O_2$ (Fig. 5). In addition, FaHOG1 and FaWC1 were required and dispensable, respectively, for osmotic stress tolerance (Fig. 6). Moreover, the recognized signature light responses of *F. asiaticum*, including light-induced carotenoid accumulation, UV damage tolerance, and perithecium formation, were examined. The results indicated that the FaHOG1 was required for carotenoid biosynthesis and perithecium development; however, overexpressing FaHOG1 in the Δ*Fawc1* mutant did not rescue its defects in these phenotypic responses to light (Fig. 9), suggesting that FaWC1 and FaHOG1 may function independently in regulating carotenoid biosynthesis and perithecium formation in *F. asiaticum*. Moreover, FaHOG1 was dispensable for UV tolerance, while overexpressing FaHOG1 did not rescue hypersensitivity of the Δ*Fawc1* mutant to UV stress (Fig. 9). Collectively, these findings lead to the assumption that FaWC1 can regulate the *Fahog1* expression level to cope with oxidative stress but not with light signal or salt stress.

**The virulence deficiency of the Δ*Fawc1* mutant can be recovered by overexpressing FaHOG1.** Considering that virulence reduction in Δ*Fawc1* is likely due to its defects in tolerance to oxidative stress, while overexpressing *Fahog1* in the Δ*Fawc1* mutant could recover its oxidative resistance, our next infection assays evaluated virulence of the WT and mutant strains. When inoculated on wheat coleoptiles, the Δ*Fawc1* and Δ*Fahog1* mutants both caused significantly smaller lesion areas than the WT strain. In contrast, the Δ*Fawc1*::FaHOG1$^{OE}$, Δ*Fahog1*-C, and Δ*Fahog1*::FaHOG1$^{OE}$

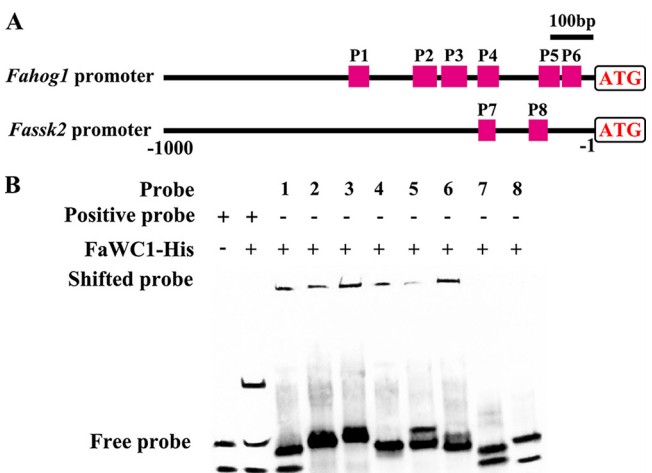

**FIG 7** EMSA. (A) Predicted binding *cis* elements of FaWC1 in the promoter regions of *Fahog1* and *Fassk2*. P1 to P8 represent the sites of probes 1 to 8 (Table S2). (B) EMSA shows that FaWC1 can bind the promoter regions of *Fahog1* but not *Fassk2*. The experiment was repeated twice independently with similar results.

strains caused lesions with severities comparable to that of lesions caused by the WT (Fig. 10A and C). Additionally, both the Δ*Fawc1* and Δ*Fahog1* mutants showed apparent defects in causing stem rot symptoms when they were inoculated on maize stems. Furthermore, the Δ*Fawc1*::FaHOG1^OE, Δ*Fahog1*-C, and Δ*Fahog1*::FaHOG1^OE strains caused levels of stem rot similar to or higher than the WT strain (Fig. 10B and D). All these data together suggest that the FaWC1 modulates virulence of *F. asiaticum* by regulating the HOG1 pathway gene expression levels.

## DISCUSSION

*F. asiaticum* is a prevalent fungal pathogen of cereal crops in east Asia, causing risks to the safety of food and feed products in this area (35). However, the understanding of the virulence mechanisms in *F. asiaticum* remains fragmentary. Light can regulate various aspects of *F. asiaticum* life, including UV damage resistance, carotenoid production, development of sexual fruiting bodies, etc. As found in other fungi with evident light responses, the *F. asiaticum* genome contains the well-conserved White Collar complex (WCC) fungal photoreceptor proteins, which are required for modulating the characterized light responses (34). Furthermore, the WCC has been recognized to regulate fungal pathogenicity in several plant pathogens, such as *Magnaporthe oryzae*, *Cercospora zeae-maydis*, and *Botrytis cinerea* (15–17, 34). Notably, our previous study also revealed that the virulence of *F. asiaticum* could be regulated by the individual WCC component FaWC1 (34). However, how the WCC orthologs are involved in regulating pathogenicity remained open to be explored on a case-by-case basis, as the

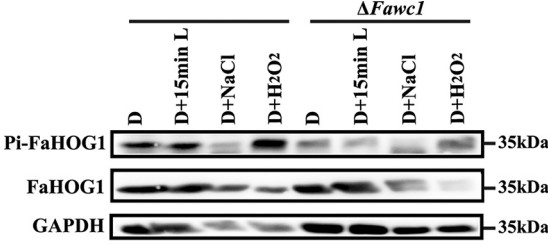

**FIG 8** Western blot assay to measure FaHOG1 phosphorylation levels in WT and Δ*Fawc1* mutant strains. Each strain was precultured for 2 days, followed by treatment with 0.5 M NaCl, 2.5 mM $H_2O_2$, or full-spectrum visible light for 15 min. Protein extraction and Western blotting were performed, and FaHOG1 and phosphorylated FaHOG1 proteins were detected using anti-Hog1 antibody (F-9; sc-365609) and phosphorylated p38 (Thr180/Tyr182) antibodies, respectively. GAPDH was detected using the anti-GAPDH antibody as a control.

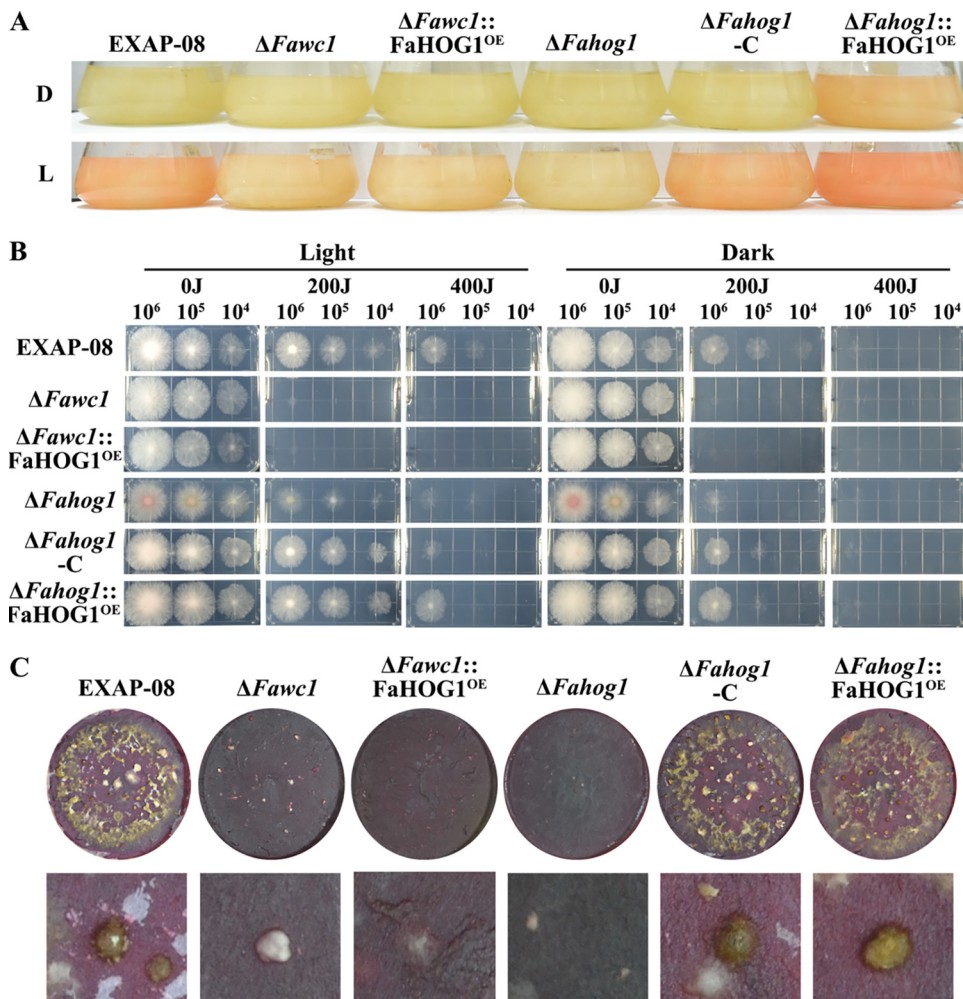

**FIG 9** Overexpression of FaHOG1 could not rescue the defects of the Δ*Fawc1* mutant in light-sensing responses. (A) Cultures of the test strains in liquid CM were incubated in darkness or light for 4 days. Both FaWC1 and FaHOG1 were required for inducing photopigmentation. (B) Serially diluted spore suspensions of the test strains were added dropwise to solid CM. After exposure to 0, 200, or 400 J UV-C radiation, the samples were cultured under light or dark conditions. Based on colony growth activities, FaHOG1 is not required for UV damage tolerance that is regulated by FaWC1. (C) Cultures of the test strains in solid carrot medium were incubated in UV light (wavelength, 365 nm) to induce perithecium development. Both FaWC1 and FaHOG1 were required for perithecium formation, but overexpression of FaHOG1 could not rescue the defect of the Δ*Fawc1* mutant in development of sexual reproduction.

functions of the light receptors varied among different fungal species, even though they definitely retain conserved light-sensing functions.

It is worth noting that the ROS formation elicited by light is considered a general phenomenon in fungi (12). Thus, the effects of light irradiance on fungal cells can be caused directly by light signaling of photoreceptors, indirectly by oxidative influence of photooxidation, or both. In addition, light sensed via fungal photoreceptors could be a warning signal of the coming oxidative stress caused by excessive light exposure and thus activate the expression of genes related to oxidative tolerance in fungi. In this context, it is reasonable that expression analysis of the light responses of the fungi *Aspergillus nidulans* and *Neurospora crassa*, which possess dedicated light-sensing mechanisms, revealed many oxidative stress-related genes to be activated by light (8, 36). Similarly, transcriptomic studies on the soil plant symbiont fungus *Trichoderma atroviride* showed that a significant proportion of the light-induced genes are related to oxidative and other types of stress responses, suggesting an overlap between photobiology and stress responses in fungi with dedicated light receptors (37, 38). Meanwhile,

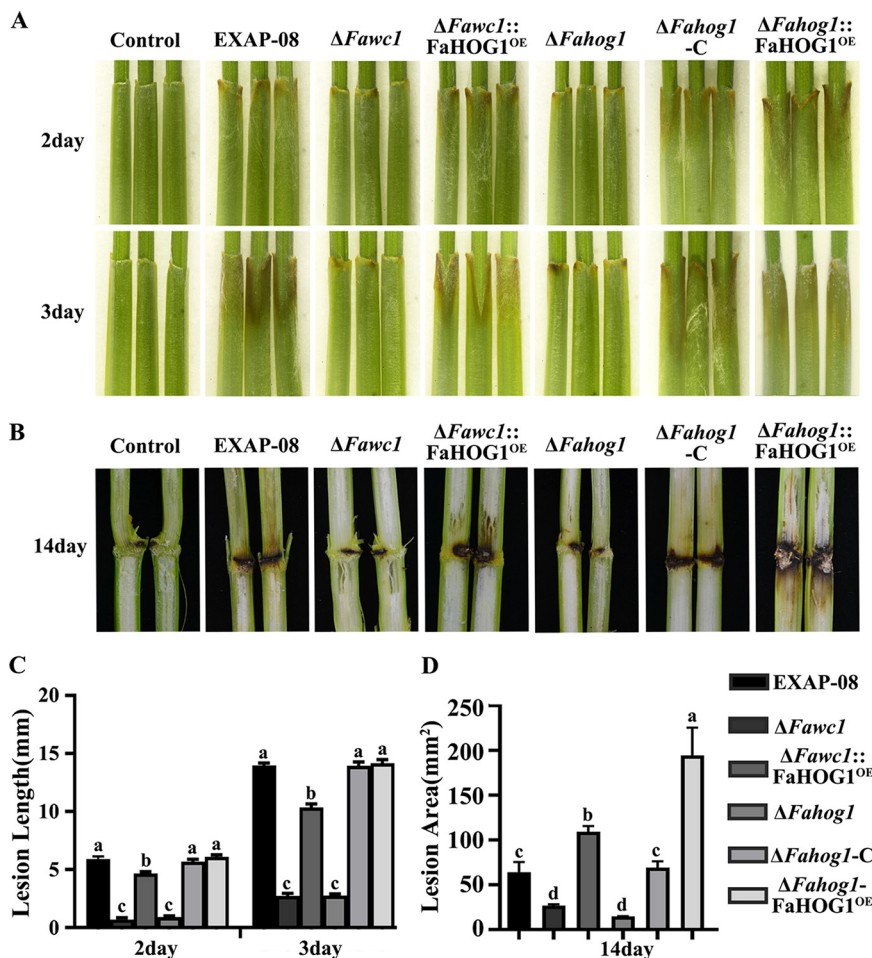

**FIG 10** The virulence deficiency of the Δ*Fawc1* mutant is recovered by overexpression of FaHOG1. When inoculated on wheat coleoptiles (A) and maize stems (B), the Δ*Fawc1* and Δ*Fahog1* mutants both caused significantly smaller lesions than the WT strain. In contrast, the Δ*Fawc1*::FaHOG1^OE, Δ*Fahog1*-C, and Δ*Fahog1*::FaHOG1^OE strains led to lesions with severities comparable to those of the WT. Images were recorded at 3 and 15 days after inoculation for wheat coleoptiles and maize stems, respectively. (C and D) Measurements of lesions caused by infection with WT, Δ*Fawc1*, Δ*Fawc1*::FaHOG1^OE, Δ*Fahog1*, Δ*Fahog1*-C, and Δ*Fahog1*::FaHOG1^OE strains on wheat coleoptiles and maize stems, respectively. Columns with different letters are significantly different from each other at a P value of <0.05.

plants have different strategies to defend against microbial infections, one of which is increasing the ROS production, also known as oxidative burst (39, 40). Consequently, successful pathogens should be able to cope with such hyperoxidative stress during interaction with their hosts (41). Interestingly, the light receptor WCC of *B. cinerea* is required to cope with the oxidative stress that is caused by excessive light exposure or arises during host invasion, and WCC is thus required for full virulence under excessive light (15). The present study additionally observed that the FaWC1, or more specifically the ZnF domain of FaWC1, is required for normal ROS scavenging activity, and the attenuated virulence of the Δ*Fawc1* mutant is accompanied by and likely due to increased accumulation of ROS, as the addition of antioxidants to the inoculation droplets restored virulence of the mutant to wild-type levels. Subsequently, it is of great importance to elucidate the oxidative stress tolerating mechanisms regulated by FaWC1, as both ROS generation and detoxification are important for the virulence of pathogenic fungal species (42).

In yeasts and filamentous fungi, the HOG MAPK pathway is well conserved for regulating response to hyperosmotic stresses (43, 44). Additionally, in several fungi, including *Cryptococcus gattii* (45), *Magnaporthe oryzae* (23), *B. cinerea* (46), and *F. graminearum* (47), the HOG pathway also regulates responses to oxidative stress. Moreover, the HOG pathway is associated with light signaling in fungi; in *A. nidulans*, the light signal perceived by

phytochromes can be transmitted from the cytoplasm using the HOG MAP kinase pathway into the nuclei (26), while in *Trichoderma guizhouense*, HOG1 plays a pivotal role in the regulation of blue-light-responsive genes (48). The present study found that the Δ*Fawc1* mutant demonstrated low expression levels of the HOG1 pathway and its downstream ROS-degrading-enzyme genes. As the WCC proteins are known as transcription factors, the altered phenotypes of the deletion mutant can be ascribed to impaired transcription patterns of the downstream genes regulated by the WCC. Consequently, FaHOG1, the core component of the HOG MAPK pathway, was overexpressed in Δ*Fahog1* and Δ*Fawc1* mutants. It was not surprising that the Δ*Fahog1* mutant showed a phenotype of hypersensitivity to osmotic and oxidative stresses, and the Δ*Fahog1*::FaHOG1[OE]-overexpressing strain recovered all these defects. However, only the oxidative tolerance and virulence defects of Δ*Fawc1* were successfully restored in the Δ*Fawc1*::FaHOG1[OE] strain, which still showed no responses to light stimulus.

One of the obvious WCC-regulated light responses in *F. asiaticum* is accumulating pigments (34), which may protect the fungal cells either by directly absorbing UV light or by scavenging the radiation-generated ROS intermediates. Photopigmentation constitutes a light response observed in many other ascomycete molds, including *N. crassa*, *Fusarium fujikuroi*, *Fusarium oxysporum*, and *Cercospora kikuchii*, as well as in the model zygomycete *Phycomyces blakesleeanus*. Moreover, the WC-1 orthologs appear to be involved in the photopigmentation responses in all species in which the functions of the photo receptors have been tested (3, 4, 12). Interestingly, loss of FaHOG1 in *F. asiaticum* also led to deficiency in the light-induced pigmentation, and the involvement of FaHOG1 in regulating pigment production seems to be dependent on the presence of FaWC1. Similarly, FaHOG1 and FaWC1 are both required for perithecium development induced by UV-A light with a peak wavelength at 365 nm, while overexpressing FaHOG1 in the Δ*Fawc1* mutant did not recover the mutant's defects in the context of sexual reproduction. In contrast, FaHOG1 and FaWC1 are dispensable and required, respectively, for light-induced UV tolerance in in *F. asiaticum*. Hence, these data collectively imply that the defects of the Δ*Fawc1* strain in light responses, including light-induced pigmentation, UV tolerance, and sexual fruiting development, should not be ascribed to the low expression of FaHOG1 in the mutant. One possible explanation for these phenomena is that FaHOG1 and FaWC1 may account for regulation of different essential components of the pathways for the pigment biosynthesis and sexual reproduction in *F. asiaticum*. The findings that some but not all phenotypes of the Δ*Fawc1* mutant can be restored by overexpressing FaHOG1 may indicate the functional versatility of the conserved WC1 protein, which possesses varied downstream targets that are independently responsible for light sensing and other stress responses.

The HOG pathway of fungi represents one central module for regulating stress responses, including oxidative stress caused by either the external environment or hostile interacting organisms. Consequently, the HOG pathway is also commonly required for host colonization by pathogenic fungi (49). In *F. graminearum*, another phylogenetically differentiated species that, like *F. asiaticum*, belongs to the *Fusarium graminearum* species complex, the expression and nucleus localization of FgHOG1 can be induced by a hyperosmotic stimulus, and the HOG cascade can orchestrate numerous physiological functions by regulating hyphal growth, development, and virulence as well as hyperosmotic and oxidative stress responses (47). As expected, the HOG1 homolog demonstrated similar functions in *F. asiaticum*. Although *in vivo* evidence for direct regulation of FaWC1 on transcription of *Fahog1* is still lacking, the assumption that appropriate responsiveness of *Fahog1* to stresses being regulated by FaWC1 is well supported by the following findings: the promoter region of *Fahog1* contains binding motifs of FaWC1, and the increases in expression, phosphorylation, and nucleus localization levels of FaHOG1 in response to oxidative stress were dependent on the presence of FaWC1. The only exception is that when the test conidia were treated with hyperosmotic stress, the expression and nucleus localization of FaHOG1 can be induced in a FaWC1-independent manner. These findings imply that the blue-light receptor FaWC1 can respond to multiple stressful stimuli rather than to light alone. The transcript abundance of the HOG

**TABLE 1** Wild-type and transgenic strains of *F. asiaticum* used in this study

| Strain or genotype | Description | Reference |
|---|---|---|
| WT (EXAP-08) | Wild type | 50 |
| ΔFawc1 | *Fawc1* deletion mutant of EXAP-08 | 34 |
| ΔFawc1-C | FaWC1 complemented transformant | 34 |
| ΔFawc1::FaWC1-GFP | FaWC1-GFP complemented transformant | This study |
| ΔFawc1-C$^{\Delta ZnF}$ | Deletion of the ZnF domain of FaWC1 | 34 |
| ΔFawc1-C$^{\Delta LOV}$ | Deletion of the LOV domain of FaWC1 | 34 |
| ΔFahog1 | *Fahog1* deletion mutant of EXAP-08 | This study |
| ΔFahog1::FaHOG1-GFP | FaHOG1-GFP complemented transformant | This study |
| ΔFawc1::FaHOG1-GFP | FaHOG1-GFP expressed in ΔFawc1 | This study |
| ΔFawc1::FaHOG1$^{OE}$ | Overexpresses FaHOG1 in ΔFawc1 | This study |
| ΔFahog1::FaHOG1$^{OE}$ | Overexpresses FaHOG1 in ΔFahog1 | This study |

pathway is controlled by FaWC1, although FaHOG1 and FaWC1 still have their own independent roles in regulating stress responses and metabolisms.

## MATERIALS AND METHODS

**Fungal strains and culture conditions.** The wild-type *Fusarium asiaticum* strain EXAP-08 and the mutants used in this study are listed in Table 1. The *Fahog1* deletion mutant was generated by transforming the knockout cassette of an overlap PCR product into protoplasts of the wild-type strain, followed by molecular characterization of the positive transformants as described previously (34). Unless indicated otherwise, the fungal strains were cultivated at 23°C in complete medium (CM) (10 g glucose, 2 g peptone, 1 g yeast extract, 1 g Casamino Acids, nitrate salts, trace elements, 0.01% vitamins, 10 g agar, and 1 L water [pH 6.5]) for mycelial growth and morphological analysis. Liquid carboxymethyl cellulose (CMC) medium was used to culture the *F. asiaticum* strains by shaking (180 rpm) for 5 days to obtain asexual conidia. The test fungal strains were cultured on carrot agar medium to observe sexual development according to a previously reported protocol (34).

**Generation of the FaWC1-GFP and FaHOG1-GFP fusion constructs and transformants.** For complementation assays, the 1.5-kb promoter regions and coding sequences without stop codons of *Fawc1* and *Fahog1* were amplified and cloned into flu6 vector using a one-step cloning kit (Yeasen, China). The resulting PNR2-*Fawc1*-GFP vector was transformed into the *Fawc1* deletion mutant to obtain the ΔFawc1::FaWC1-GFP transgenic strain. Protoplast preparation and polyethylene glycol (PEG)-mediated transformation of *F. asiaticum* were performed as described elsewhere (34). For selection of transformants, Geneticin was added to a final concentration of 50 μg/mL to the medium. Similarly, PNR2-*Fahog1*-GFP vector was transformed into either *Fawc1* or *Fahog1* deletion mutants, yielding the ΔFawc1::FaHOG1-GFP and ΔFahog1::FaHOG1-GFP transgenic strains, respectively. The transformants expressing the FgWC1-GFP or FgHOG1-GFP construct were phenotypically analyzed in comparison with the recipient strains to confirm that they had achieved functional complementation. Additionally, the GFP signals in conidia and germlings were observed with a fluorescence microscope (Axio Imager Z2; Zeiss, Germany), using its light, fluorescence, and light/fluorescence-merged fields. A neutral-density filter set, D/A (d = 25), was used during microscopic analysis. The excitation and emission wavelengths used were 488 nm and 500 to 550 nm for GFP.

**Assay for nucleus staining.** To stain the nuclei, the germinating spores of the wild-type and mutant strains of *F. asiaticum* on glass slides were stained with 4′,6-diamidino-2-phenylindole (DAPI) at 0.5 mg mL$^{-1}$ for 10 min before being analyzed under the Zeiss microscope with the filter sets for excitation at 358 to 360 nm and emission wavelength detection from 460 to 461 nm.

**Assays for light-responsive phenotypic analysis.** Carotenoid biosynthesis, photorepair of UV damage, and the development of the perithecia (sexual reproductive structures) are characteristic light responses in *F. asiaticum*. The wild-type and mutant strains were tested for these light-responsive phenotypes according to the reported procedures (34).

**Expression analysis by qRT-PCR.** In order to study the expression of target genes in *F. asiaticum*, 200-μL aliquots of conidial suspensions (10$^6$/mL) of test strains were inoculated on solid CM with cellophane overlays. After incubation at 23°C for 12 h, the samples were treated with either light or stress agents for 1 h. Subsequently, the mycelium samples were harvested followed by immediate freezing in liquid nitrogen. The frozen mycelium samples were subjected to RNA extraction by using Qiagen reagent (Germany). One microgram of each RNA sample was used for reverse transcription with the reverse transcription (RT) reagent kit (Perfect Real Time) (TaKaRa Biotechnology Co., Dalian, China) according to the manufacturer's instructions. The resulting single-stranded cDNA was later used as a template for quantitative reverse transcription-PCR (qRT-PCR) in a CFX96TM real-time system (Bio-Rad, Inc., USA) using TaKaRa SYBR Premix Ex Taq (TaKaRa Biotechnology, Co., Dalian, China). Transcript levels of the target genes were normalized against tubulin gene expression. The experiment was repeated three times with triplicate samples. The gene expression levels were calculated using the $2^{-\Delta\Delta CT}$ method. All primers used in the present study are listed in Table S1.

**Stress sensitivity assays.** For conidial germination assays, 20-μL droplets of suspensions containing 10$^5$ conidia per mL were applied dropwise to glass slides. After incubation at 23°C in a humidified chamber for 2 h, the germinating conidia were observed under a light microscope. For the colony growth assay, the conidial suspension was added dropwise to CM to test colony expansion growth. For testing

sensitivities to various stresses, the conidial suspensions on the glass slides or the CM were supplemented with 0.5 M NaCl, 0.5 M KCl, or 2.5 mM $H_2O_2$.

**DAB staining of fungal colonies.** The wild-type and mutant strains were inoculated on CM in 12-well culture plates. After culture for 1.5 days, 2 mL of a 0.1% DAB solution was added to each colony. After overnight staining, the supernatant staining solution was pipetted into a 1.5-mL test tube. The remaining colonies and the collected staining solutions were photographed and compared for DAB staining intensity.

**EMSA.** *Fawc1* was amplified and cloned into the pET32a vector to generate the FaWC1-His fusion construct. The resulting construct was transformed into *Escherichia coli* strain BL21. The recombinant FaWC1-His protein expressed in BL21 was purified with Ni-Sepharose beads. Probe DNAs labeled with biotin and their reverse complementary chains were synthesized. EMSAs were performed with a LightShift chemiluminescent EMSA kit (Thermo Scientific, USA) according to the manufacturer's instructions. Briefly, purified FaWC1-His and biotin-labeled probes (Table S2) were incubated in a 20-$\mu$L reaction mixture for 20 min at 15°C. The reaction products were electrophoresed on 6% polyacrylamide gels at 95 V on ice for about 60 min and transferred to a positively charged nylon membrane. Signals were detected by Chemidoc mp (Bio-Rad, USA). The experiment was performed three times.

**Western-blot analysis.** The fungal strains for test were inoculated on CM with cellophane overlays. After preculture for 2 days, the mycelia on cellophane were treated with 0.5 M NaCl, 2.5 mM $H_2O_2$, or full-spectrum visible light for 15 min. Subsequently, the mycelial samples were harvested and subjected to standard procedures for protein extraction and Western blotting. FaHOG1 and phosphorylated FaHOG1 proteins were detected using anti-Hog1 antibody (F-9; sc-365609) and phosphorylated p38 (Thr180/Tyr182) antibodies, respectively. Glyceraldehyde-3-phosphate dehydrogenase (GAPDH) was detected using the anti-GAPDH antibody as a control.

**Virulence assays on wheat and maize.** Fungal conidia were harvested from CMC medium and suspended in water. The concentration was adjusted to $10^6$ conidia/mL. Coleoptiles of wheat (*Triticum aestivum* cultivar Zhongyuan 98-68) grown for 2 days were cut with scissors and inoculated with 3 $\mu$L conidial suspension, and then they were kept in a transparent box at 23°C under a 12 h/12 h light cycle and humid conditions. The infected samples were photographed, and the lesion areas were calculated using ImageJ software.

To carry out pathogenicity assay with maize stems, the *Zea mays* inbred line B73 was grown in a greenhouse which was maintained at 26°C to obtain 10-leaf seedlings. The stems were punctured with toothpicks dipped in conidial suspensions. Symptom development was observed after splitting the stalks along the inoculation sites at 14 days postinoculation (dpi). The areas of lesions in the infected stems were calculated following previously reported methods (47).

**Statistical analysis.** The data obtained in this study were analyzed with analysis of variance (ANOVA) followed by Duncan's multiple range tests ($P < 0.05$) for comparison of means by using SPSS 17.0.

## SUPPLEMENTAL MATERIAL

Supplemental material is available online only.
**SUPPLEMENTAL FILE 1**, PDF file, 0.2 MB.

## ACKNOWLEDGMENTS

This study was financially supported by the National Natural Science Foundation of China (no. 32061133006 and 31972121) and by the Natural Science Foundation of Shanghai (no. 21ZR1421600), China.

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
