## [Reviewer comments · Microbiology Spectrum]

Microbiology Spectrum

White Collar 1 modulates oxidative sensitivity and virulence via regulating the HOG1 pathway in *Fusarium asiaticum*

Ying Tang, Yan Tang, Dandan Ren, Congcong Wang, Yao Qu, Li Huang, Yongjun Xue, Yina Jiang, Yiwen Wang, Ling Xu, and Pinkuan Zhu

Corresponding Author(s): Pinkuan Zhu, East China Normal University

Review Timeline:

Submission Date:	December 18, 2022
Editorial Decision:	February 4, 2023
Revision Received:	April 3, 2023
Editorial Decision:	April 17, 2023
Revision Received:	April 19, 2023
Accepted:	April 21, 2023

Editor: Kaustuv Sanyal

Reviewer(s): The reviewers have opted to remain anonymous.

Transaction Report:

DOI: <https://doi.org/10.1128/spectrum.05206-22>

February 4, 2023

Dr. Pinkuan Zhu
East China Normal University
School of Life Sciences
500 Dongchuan Road
Shanghai 200241
China

Re: Spectrum05206-22 (White Collar 1 modulates oxidative sensitivity and virulence via regulating the HOG1 pathway in *Fusarium asiaticum*)

Dear Dr. Pinkuan Zhu:

Thank you for submitting your manuscript to Microbiology Spectrum. As you see from the comments appended below while both reviewers find the work important, the second reviewer suggested some additional experimental evidence to support the conclusion. When submitting the revised version of your paper, please provide (1) point-by-point responses to the issues raised by the reviewers as file type "Response to Reviewers," not in your cover letter, and (2) a PDF file that indicates the changes from the original submission (by highlighting or underlining the changes) as file type "Marked Up Manuscript - For Review Only". Please use this link to submit your revised manuscript - we strongly recommend that you submit your paper within the next 60 days or reach out to me. Detailed instructions on submitting your revised paper are below.

Link Not Available

Sincerely,

Kaustuv Sanyal

Journals Department
Reviewer comments:

Reviewer #1 (Comments for the Author):

The research defines the basis by which a blue light sensor impacts the pathogenicity of the plant pathogen *Fusarium asiaticum*, a close relative of *F. graminearum*, and a major problem in east Asia. Some of the authors had previously established the key part of FaWC1 for pathogenicity is its DNA binding zinc finger domain. Here they discover the downstream components involved, tied to a decrease in oxidative stress tolerance as mediated via the HOG signalling pathway. The findings should be of wide interest to those exploring signal transduction pathways in fungi. First, for many years it was puzzling why a light receptor

would be involved in pathogenicity in different fungi, and here is another example showing that the role of this highly conserved protein is independent of the role in light sensing. Second, there are previous links between light sensing and the HOG pathway (perhaps the best example being in *Aspergillus nidulans* and red light sensing phytochrome), so this work nicely establishes a link between the blue light pathway and HOG pathway.

Two minor questions for consideration are:

Mutation of HOG1 often causes a suite of phenotypes in addition to increased sensitivity to oxidative stress. Are any of these seen in the FaHOG1 mutant, and, to follow on if they are, also in the FaWC1 mutant?

The current model has FaWC1 controlling transcription of FaHOG1; it might be good to look for DNA binding sites in the promoter as another line of evidence for direct regulation.

Minor typographical or editorial considerations.

Line 44: 'FaWC1 relies on its'.

Line 47: 'It's' to 'It was'.

Line 50: is due to a defect in ROS'.

Lines 61-62: 'but how WCC determines fungal pathogenicity'.

Line 67: 'pathogenicity in an important'.

Line 93: 'could' to 'can'.

Lines 111, 363: 'It's' to 'It is'.

Line 134: move 'could' to be 'components could be associated'.

Line 141: 'predominant set of pathogens for'.

Line 151: 'demonstrated a light-independent role'.

Line 183: 'glycol' for 'glycerol'.

Line 214: clarify 'The experiment was repeated three times', i.e. does this mean as written done on three independent occasions, or does it refer to biological replicates?

Line 215: 'used in the present study'.

Line 226: 'cultivar'.

Line 227: 'cut with scissors and'.

Lines 231-2: grown in a greenhouse'.

Line 241: result comes too early, so perhaps 'FaWC1 has a role in oxidative stress tolerance'.

Line 250: is the DAB staining information in the Materials and Methods?

Line 272: might be 'as the light illumination continued,'.

Line 279: 'To test the hypothesis that FaWC1...'.

Lines 293, 390, 414: change 'dwarf' to something like 'low'.

Line 318: 'testify' to 'test'.

Line 349-50: 'in east Asia, causing risks to the...'.

Line 356: 'have' to 'has'.

Line 359: 'how the WCC orthologs are involved'.

Line 370: 'defend'.

Line 373: delete 'it's revealed that'.

Line 381: 'were proved to be' to 'are'.

Line 393: may be 'impaired' in place of 'frustrated'.

Line 393-395: elaborate on how this increased expression was shown in the mutant.

Lines 402, 509: 'Photo-pigmentation' for consistency elsewhere in the text.

Line 403: 'oxysporum.'

Line 424: 'belongs to the same FGSC'.

Line 428: 'expected'.

Line 435: delete 'the' to be 'although FaHOG1 and FaWC1'.

Line 451: space '1 mM'.

Line 461: delete 'feeding'.

Reviewer #2 (Comments for the Author):

The white collar complex (WCC) is the main photoreceptor for fungal responses to blue light. WC1 also has light-independent functions. Having shown previously that the virulence and light-sensing functions can be separated, this report brings evidence that WC1 of *Fusarium asiaticum* (closely related to *F. graminearum*) promotes virulence through its role in sensing oxidative stress, independently of the flavin-binding LOV domain.

Two main issues would need to be addressed, both related to the WC1-Hog1 genetic interaction. First, rescue of *wc1* phenotypes by Hog1 overexpression supports the hypothesis that the Hog1 MAPK may be acting downstream of WC1, or co-

regulating some of the same genes. The critical factor though for MAP kinases is their phosphorylation level, not necessarily their expression level. Second, although much evidence is gained from a line overexpressing Hog1, no data are provided to conclude that Hog1 is overexpressed. These concerns could be addressed by immunoblot assays; for overexpression, strong support could be obtained, at least, by qPCR showing increased expression levels of Hog1 in the overexpression line. The qPCR assays are already in place (Figure 4) but it seems the OE line was not yet tested.

Other comments:

major: in Figures 2 and 5, only single spore images are shown. More images, or quantitation would be needed to strengthen the conclusions that GFP nuclear retention is changed by the treatments.

minor:

line 199 and elsewhere: "testify" better "tested"

line 184 - 50 mg/ml - probably micrograms/ml?

lines 197-198 there seems to be a typo in the wavelengths, DAPI is detected by UVA excitation and blue fluorescence emission

line 200 - better: photorepair of UV damage

line 247 - FaWC2 should be FaWC1, if I understood correctly

line 246-248 Growth rate inhibition is stronger in the wc1 mutant. This implies that WCC is not the photoreceptor for this particular light effect, an interesting finding but not discussed either here or in the Discussion.

line 255 - sensitivity to ROS or ROS production? Figure 2A looks like ROS production.

line 277 typo into to

line 238 - all these four - the pattern differs between the different genes, please give a little more detail

line 293 (and in another instance as well) "dwarf expression" better "decreased expression"

line 344-5 can replace "Meanwhile" with "Futhermore" and also rewrite for example as follows - ..strains caused levels of stem rot similar to or higher than the WT strain (Figure 8).

line 366-7 Similarly, transcriptomic studies on ...

line 386: the HOG pathway is associated with light signaling ...

line 393 "Frustrated" replace with "Altered" or "Defective"

line 409 and elsewhere - black light is an outdated term for UV; replace with UV and state which UV band (UVA, UVB, UVC)

line 414 and elsewhere "dwarf expression" replace "decreased expression" or under-expression

line 425 - FGSC (Fungal Genetics Stock Center?) if so, not clear what this means

line 428 as expected,

line 433 can respond to

line 433 stimuli, rather than to light alone.

line 638 (references) delete "dagger"

Staff Comments:

Preparing Revision Guidelines

Please return the manuscript within 60 days; if you cannot complete the modification within this time period, please contact me. If you do not wish to modify the manuscript and prefer to submit it to another journal, please notify me of your decision immediately so that the manuscript may be formally withdrawn from consideration by Microbiology Spectrum.

Dear Editors and Reviewers,

Thank you very much for your message dated on 4th Feb 2023 with very constructive comments and suggestions concerning our manuscript entitled “White Collar 1 modulates oxidative sensitivity and virulence via regulating the HOG1 pathway in *Fusarium asiaticum*” (Spectrum05206-22). Those comments are helpful for revising and improving our paper, and the kind suggestions are also of crucial significance to our future research.

We have carefully studied all the comments, carried out additional experiments as suggested, and made revisions in the updated manuscript, which we hope to satisfactorily meet with approval. All specific revisions have been marked in red font in the revised manuscript, and the detailed explanations are addressed in the following Point-by-Point Responses to Reviewers' Comments.

Point-by-Point Responses to Reviewers' Comments

-Reviewer #1:

The research defines the basis by which a blue light sensor impacts the pathogenicity of the plant pathogen *Fusarium asiaticum*, a close relative of *F. graminearum*, and a major problem in east Asia. Some of the authors had previously established the key part of FaWC1 for pathogenicity is its DNA binding zinc finger domain. Here they discover the downstream components involved, tied to a decrease in oxidative stress tolerance as mediated via the HOG signaling pathway. The findings should be of wide interest to those exploring signal transduction pathways in fungi. First, for many years it was puzzling why a light receptor would be involved in pathogenicity in different fungi, and here is another example showing that the role of this highly conserved protein is independent of the role in light sensing. Second, there are previous links between light sensing and the HOG pathway (perhaps the best example being in *Aspergillus nidulans* and red light sensing phytochrome), so this work nicely establishes a link between the blue light pathway and HOG pathway.

Response: Thanks for the reviewer’s general comments on this manuscript.

Two minor questions for consideration are:

Mutation of HOG1 often causes a suite of phenotypes in addition to increased sensitivity to oxidative stress. Are any of these seen in the FaHOG1 mutant, and, to follow on if they are, also in the FaWC1 mutant?

Response: Thanks very much for this comment. The *Fahog1* knock-out mutant indeed shows a suit of defects, including but not limited to hyphal growth, resistance to oxidative agent, osmotic stress tolerance, sexual and asexual development, carotenoid synthesis and pathogenicity. Some of the Δ *Fahog1* defects, including resistance to oxidative agent, carotenoid synthesis, sexual development, and pathogenicity, were also seen in the Δ *Fawc1* mutant. Overexpressing *Fahog1* in the Δ *Fawc1* mutant could only restore the defects in resistance to oxidative agent and pathogenicity, but not the defects in sexual development and carotenoid synthesis. These findings may suggest that the transcript abundancy of *Fahog1* pathway is regulated by FaWC1 in response to oxidative stimulus and host infection, although the FaHOG1 and FaWC1 can still have their own independent roles in regulating stress responses and metabolisms.

The current model has FaWC1 controlling transcription of FaHOG1; it might be good to look for DNA binding sites in the promoter as another line of evidence for direct regulation.

Response: Thanks for the constructive comment. We fully agree that it is worth to explore whether direct regulation exist between FaWC1 and the Hog1 pathway genes. Actually, we did try ChIP assay with the Δ *Fawc1*::FaWC1-GFP strain, however, we didn't succeed in obtaining the immunoprecipitated DNA library using the GFP antibody, probably because the absolute expression abundancy of the FaWC1-GFP protein is low in the fungus. Alternatively, we carried out the in vitro strategy, Electrophoretic mobility shift assay (EMSA). There are two predicted motifs, TCTTCCTCCTC and CCACTCTAT, in the *Fahog1* promoter region that can be bound by FaWC1. Subsequently, EMSA assay confirmed that FaWC1 could indeed bind to the promoter regions of *Fahog1* (Figure 7 in the revised manuscript). Although, in vivo evidence for direct regulation of FaWC1 on transcription of *Fahog1* is still lacking, the genetic and in vitro biochemical data obtained in this study could support the assumption that appropriate expression patterns of *Fahog1* in response to stresses is regulated by FaWC1.

Minor typographical or editorial considerations.

Line 44: "FaWC1 relies on its".

Response: It has been revised as suggested.

Line 47: "It's' to 'It was".

Response: It has been revised as suggested.

Line 50: “is due to a defect in ROS”.

Response: It has been revised as suggested.

Lines 61-62: “but how WCC determines fungal pathogenicity”.

Response: It has been revised as suggested.

Line 67: “pathogenicity in an important”.

Response: It has been revised as suggested.

Line 93: “could' to 'can”.

Response: It has been revised as suggested.

Lines 111, 363: “It's' to 'It is”.

Response: It has been revised as suggested.

Line 134: move “could” to be “components could be associated”.

Response: It has been revised as suggested.

Line 141: “predominant set of pathogens for”.

Response: It has been revised as suggested.

Line 151: “demonstrated a light-independent role”.

Response: It has been revised as suggested.

Line 183: “glycol” for “glycerol”

Response: Thanks for this comment. It has been revised as suggested.

Line 214: clarify 'The experiment was repeated three times', i.e. does this mean as written done on three independent occasions, or does it refer to biological replicates?

Response: Thank you very much for this comment. Since qRT-PCR is highly sensitive for measuring gene expression levels, the qRT-PCR experiments in this study were repeated three independent occasions with triplicate samples in each repeat. Accordingly, the method is clarified as “The experiment was repeated three times with triplicate samples”.

Line 215: “used in the present study”.

Response: It has been revised as suggested.

Line 226: “cultivar”.

Response: It has been revised as suggested.

Line 227: “cut with scissors and”.

Response: It has been revised as suggested.

Lines 231-2: “grown in a greenhouse”.

Response: It has been revised as suggested.

Line 241: result comes too early, so perhaps 'FaWC1 has a role in oxidative stress

tolerance'.

Response: Thanks for the comment. We have revised it as suggested.

Line 250: is the DAB staining information in the Materials and Methods?

Response: Thanks for the comment. The DAB staining method has been added to the revised manuscript.

Line 272: might be “as the light illumination continued,”.

Response: It has been revised as suggested.

Line 279: “To test the hypothesis that FaWC1...”

Response: It has been revised as suggested.

Lines 293, 390, 414: change “dwarf” to something like “low”.

Response: It has been revised as suggested.

Line 318: “testify” to “test”.

Response: It has been revised as suggested.

Line 349-50: “in east Asia, causing risks to the...”.

Response: It has been revised as suggested.

Line 356: “have” to “has”.

Response: It has been revised as suggested.

Line 359: “how the WCC orthologs are involved”.

Response: It has been revised as suggested.

Line 370: “defend”.

Response: It has been revised as suggested.

Line 373: delete “it's revealed that”.

Response: It has been revised as suggested.

Line 381: “were proved to be” to “are”.

Response: It has been revised as suggested.

Line 393: may be 'impaired' in place of 'frustrated'.

Response: Thanks for the helpful comments. It has been revised as suggested.

Line 393-395: elaborate on how this increased expression was shown in the mutant.

Response: Thanks for the comment. This expression levels of *Fahog1* in $\Delta\text{Fawc1}::\text{FaHOG1}^{\text{OE}}$ strains were shown in the supplementary Figure S1 of the revised manuscript.

Lines 402, 509: “Photo-pigmentation” for consistency elsewhere in the text.

Response: Thanks for the helpful comment. We have revised the descriptions in the text as suggested.

Line 403: “oxysporum”.

Response: It has been revised as suggested.

Line 424: “belongs to the same FGSC”.

Response: It has been revised as suggested.

Line 428: “expected”.

Response: It has been revised as suggested.

Line 435: delete “the” to be “although FaHOG1 and FaWC1”.

Response: It has been revised as suggested.

Line 451: space '1 mM'.

Response: It has been revised as suggested.

Line 461: delete “feeding”.

Response: It has been revised as suggested.

Reviewer #2:

The white collar complex (WCC) is the main photoreceptor for fungal responses to blue light. WC1 also has light-independent functions. Having shown previously that the virulence and light-sensing functions can be separated, this report brings evidence that WC1 of *Fusarium asiaticum* (closely related to *F. graminearum*) promotes virulence through its role in sensing oxidative stress, independently of the flavin-binding LOV domain.

Two main issues would need to be addressed. First, rescue of *wc1* phenotypes by *Hog1* overexpression supports the hypothesis that the *Hog1* MAPK may be acting downstream of WC1, or co-regulating some of the same genes. The critical factor though for MAP kinases is their phosphorylation level, not necessarily their expression level.

Response: Thank you very much for this comment. We totally agree that a critical factor for MAP kinases is their phosphorylation level. Additional Western Blot assay was thus carried out, using the anti-*Hog1p* and anti-phospho-p38 antibodies. The result showed that loss of FaWC1 could also alter the phosphorylation level of FaHOG1 in response to stresses (Figure 8 in the revised manuscript).

Second, although much evidence is gained from a line overexpressing *Hog1*, no data are provided to conclude that *Hog1* is overexpressed. These concerns could be addressed by immunoblot assays; for overexpression, strong support could be obtained, at least, by qPCR showing increased expression levels of *Hog1* in the

overexpression line. The qPCR assays are already in place (Figure 4) but it seems the OE line was not yet tested.

Response: Thanks for the suggestion. We have detected *Fahog1* expression levels in the $\Delta Fawc1::Fahog1^{OE}$ by qPCR assays, and added the data in the revised manuscript as a supplementary Figure S1.

In Figures 2 and 5, only single spore images are shown. More images, or quantitation would be needed to strengthen the conclusions that GFP nuclear retention is changed by the treatments.

Response: Thank you very much for this important comment. We have added less magnified figures showing more spores in each view as supplementary Figure S2 to indicate the overall GFP nuclear retention status of each sample.

4. Line 199 and elsewhere: "testify" better "tested"

Response: It has been revised as suggested.

5. Line 184 - 50 mg/ml - probably micrograms/ml?

Response: Thanks for this comment. Actually, 50 mg/ml is the concentration of the geneticin mother solution, the working concentration in selection medium is 50 μ g/ml. We have revised it accordingly.

Lines 197-198 there seems to be a typo in the wavelengths, DAPI is detected by UVA excitation and blue fluorescence emission

Response: Thank you very much for your comment. I am sorry that it is a typo in the wavelengths. The Zeiss microscope with the filter set for excitation of DAPI ranges from 358-360 nm, and emission spectrum is 460-461 nm. Accordingly, the wavelengths for detecting DAPI have been revised in the text.

Line 200 - better: photorepair of UV damage

Response: Thanks for the helpful comment. We have revised it as suggested.

Line 247 - FaWC2 should be FaWC1, if I understood correctly

Response: Thank you very much for the comment. We have corrected it.

Line 246-248 Growth rate inhibition is stronger in the *wc1* mutant. This implies that WCC is not the photoreceptor for this particular light effect, an interesting finding but not discussed either here or in the Discussion.

Response: Thank you very much for this comment. The related discussion has thus been revised accordingly in the re-submitted manuscript (Line 405-409).

Line 255 - sensitivity to ROS or ROS production? Figure 2A looks like ROS production.

Response: Thanks for the comment. The sentence has been revised accordingly.

Line 277 typo into to

Response: It has been revised as suggested.

Line 283 - all these four - the pattern differs between the different genes, please give a little more detail

Response: Thank you very much for this comment. We have added more details describing the results in the revised manuscript (Line 310-316).

Line 293 (and in another instance as well) "dwarf expression" better "decreased expression".

Response: Thanks for the suggestion. We have revised them in the text.

Line 344-5 can replace "Meanwhile" with "Furthermore" and also rewrite for example as follows - . Strains caused levels of stem rot similar to or higher than the WT strain (Figure 8).

Response: It has been revised as suggested.

Line 366-7 Similarly, transcriptomic studies on.

Response: It has been revised as suggested.

Line 386: the HOG pathway is associated with light signaling ...

Response: It has been revised as suggested.

Line 393 "Frustrated" replace with "Altered" or "Defective"

Response: Thanks for the helpful comments. We have modified the related parts.

Line 409 and elsewhere - black light is an outdated term for UV; replace with UV and state which UV band (UVA, UVB, UVC)

Response: Thanks for the helpful comments. The spectrum being applied is UVA band with peak λ at 365 nm. Accordingly, the term has been revised following the suggestion.

Line 414 and elsewhere "dwarf expression" replace "decreased expression" or under-expression

Response: Thanks for the helpful comment. We have revised the descriptions in the text accordingly.

Line 425 - FGSC (Fungal Genetics Stock Center?) if so, not clear what this means

Response: Thanks for the comment. FGSC here means *Fusarium graminearum* species complex. It has been specified in the text of the revised manuscript.

Line 428 as expected,

Response: It has been revised as suggested.

Line 433 can respond to.

Response: It has been revised as suggested.

Line 433 stimuli, rather than to light alone.

Response: It has been revised as suggested.

Line 638 (references) delete "dagger".

Response: It has been revised as suggested.

April 17, 2023

Dr. Pinkuan Zhu
East China Normal University
School of Life Sciences
500 Dongchuan Road
Shanghai 200241
China

Re: Spectrum05206-22R1 (White Collar 1 modulates oxidative sensitivity and virulence via regulating the HOG1 pathway in *Fusarium asiaticum*)

Dear Dr. Pinkuan Zhu:

Thank you for submitting your manuscript to Microbiology Spectrum. There are a few minor issues to be addressed as indicated by the reviewers before the manuscript can be formally accepted for publication. As you will see your paper is very close to acceptance. Please modify the manuscript as recommended. As these revisions are quite minor, I expect that you should be able to turn in the revised paper in less than 30 days, if not sooner. If your manuscript was reviewed, you will find the reviewers' comments below.

When submitting the revised version of your paper, please provide (1) point-by-point responses to the issues raised by the reviewers as file type "Response to Reviewers," not in your cover letter, and (2) a PDF file that indicates the changes from the original submission (by highlighting or underlining the changes) as file type "Marked Up Manuscript - For Review Only". Please use this link to submit your revised manuscript. Detailed instructions on submitting your revised paper are below.

Link Not Available

Sincerely,

Kaustuv Sanyal

Reviewer comments:

Reviewer #1 (Comments for the Author):

The authors have gone above and beyond to address the reviewers' comments, including performing new experiments such as gel mobility shifts and western blotting.

There are a few minor edits still to consider.

(1) There is no information in the Materials and Methods about the western blotting.

Minor typographical points are:

Line 340: 'were' for 'was'.

Line 349: 'did not' for 'didn't'.

Line 603: add italics on '*Aspergillus nidulans*'.

Line 698: b in 'Blot' can be lower case.

Figure 7B: spelling 'Positive'.

Reviewer #2 (Comments for the Author):

This study provides evidence that the photoreceptor WC1 of *Fusarium asiaticum*, an important pathogen, promotes virulence through its role in sensing oxidative stress, independently of the flavin-binding LOV domain.

The revised version now provides good support for overexpression of Fa Hog1 (Figure S2). The immunoblot (new Figure 8) shows Hog1 phosphorylation in response to oxidative stress.

I noticed the following upon rereading:

Both the immunofluorescence and immunoblot data seem consistent with the main effect of Wc1 loss on Hog1 being, simply, decreased expression, rather than loss of induction by stress. This is an alternative explanation for the authors' consideration. Quantitation of the phospho-MAPK signal on the western blot (blots?) relative to total Hog1 and/or GAPDH signals might exclude this explanation.

Please check the new Figure numbering (noticed one reference to Figure 8 which would now be Figure 9).

Typo in the y-axis of Figure 9 (lesion)

Preparing Revision Guidelines

Please return the manuscript within 60 days; if you cannot complete the modification within this time period, please contact me. If you do not wish to modify the manuscript and prefer to submit it to another journal, please notify me of your decision immediately so that the manuscript may be formally withdrawn from consideration by Microbiology Spectrum.

Point-by-Point Responses to Reviewers' Comments

Reviewer #1:

The authors have gone above and beyond to address the reviewers' comments, including performing new experiments such as gel mobility shifts and western blotting.

There are a few minor edits still to consider.

(1) There is no information in the Materials and Methods about the western blotting.

Response: thanks for this comment. The information about the Western blot assay has been added in the revised manuscript.

Minor typographical points are:

Line 340: 'were' for 'was'.

Response: It has been revised as suggested.

Line 349: 'did not' for 'didn't'.

Response: It has been revised as suggested.

Line 603: add italics on '*Aspergillus nidulans*'.

Response: It has been revised as suggested.

Line 698: b in 'Blot' can be lower case.

Response: It has been revised as suggested.

Figure 7B: spelling 'Positive'.

Response: It has been revised as suggested.

Reviewer #2:

This study provides evidence that the photoreceptor WC1 of *Fusarium asiaticum*, an important pathogen, promotes virulence through its role in sensing oxidative stress, independently of the flavin-binding LOV domain.

The revised version now provides good support for overexpression of FaHog1 (Figure S2). The immunoblot (new Figure 8) shows Hog1 phosphorylation in response to oxidative stress.

Response: Thanks for the reviewer's general comments on this manuscript.

I noticed the following upon rereading:

Both the immunofluorescence and immunoblot data seem consistent with the main effect of Wc1 loss on Hog1 being, simply, decreased expression, rather than loss of induction by stress. This is an alternative explanation for the authors' consideration. Quantitation of the phospho-MAPK signal on the western blot (blots?) relative to total Hog1 and/or GAPDH signals might exclude this explanation.

Response: Thanks for this kind and instructive comment. We have repeated the western blot assay for several times, and each repeat demonstrated similar trend as shown in Figure 8. We tried to follow the reviewer's comment to quantify the phospho-MAPK signal on the western blots relative to total Hog1 and/or GAPDH signals via measuring grayish values of the bands. However, we found such operation was difficult, especially when defining the areas of the bands, as it could always happen that some neighbor bands overlapped a little bit. So, we prefer proposing that the main effect of FaWC1 loss on Hog1 is decreasing its expression, but we could not exclude the possibility that FaWC1 may regulate protein activity of FaHOG1 after transcription.

Please check the new Figure numbering (noticed one reference to Figure 8 which would now be Figure 9).

Response: Thanks for the careful review. The error reference to Figure 8 has been revised correctly in the updated version of manuscript.

Typo in the y-axis of Figure 9 (lesion)

Response: It has been revised as suggested.

April 21, 2023

Dr. Pinkuan Zhu
East China Normal University
School of Life Sciences
500 Dongchuan Road
Shanghai 200241
China

Re: Spectrum05206-22R2 (White Collar 1 modulates oxidative sensitivity and virulence via regulating the HOG1 pathway in *Fusarium asiaticum*)

Dear Dr. Pinkuan Zhu:

Your manuscript has been accepted, and I am forwarding it to the ASM Journals Department for publication. You will be notified when your proofs are ready to be viewed.

Sincerely,

Kaustuv Sanyal
Editor, Microbiology Spectrum
